# Self-Attention-Based Deep Learning for Missing Sensor Data Imputation in Real-Time Probe Card Monitoring

**DOI:** 10.3390/s25237194

**Published:** 2025-11-25

**Authors:** Mehdi Bejani, Marco Mauri, Stefano Mariani

**Affiliations:** 1Department of Civil and Environmental Engineering, Politecnico di Milano, 20133 Milano, Italy; stefano.mariani@polimi.it; 2Technoprobe, 23870 Cernusco Lombardone, Italy; marco.mauri@technoprobe.com

**Keywords:** BRITS, imputation, missing data, probe card, real-time monitoring, SAITS, Self-Attention-based Imputation for Time Series, sensor data

## Abstract

In industrial monitoring of semiconductor probe cards, real-time sensor data acquisition and processing are essential for anomaly detection and predictive maintenance. However, missing data resulting from possible sensor malfunctions present a significant challenge, compromising the integrity of subsequent analyses. The present study addresses this issue by applying and evaluating a state-of-the-art deep learning approach, the Self-Attention-based Imputation for Time Series model, to reconstruct corrupted signals from an industrial sensor network comprising accelerometers and microphones. A rigorous evaluation was conducted against traditional imputation methods and a powerful deep learning comparison method, the Bidirectional Recurrent Imputation for Time Series model, using a comprehensive set of time- and frequency-domain metrics. The results demonstrate that the self-attention model achieves competitive or superior accuracy, with an average improvement of 66% (with values ranging between 25% and 88%) in Mean Absolute Error over traditional methods especially in scenarios with extensive data loss, ensuring high fidelity in the reconstructed signals. The proposed analysis shows that the attention-based architecture offers a substantial practical advantage, completing training per epoch more than twenty times faster than the recurrent-based comparison method. This balance of high performance and computational efficiency makes the self-attention framework a robust and pragmatic solution to achieve data integrity in demanding monitoring and management systems.

## 1. Introduction

The advent of Industry 4.0 has led to an increased reliance on sensor networks for real-time monitoring and predictive maintenance in various industrial sectors, including semiconductor manufacturing [1]. Probe cards (PCs) are critical components in Electrical Wafer Sort (EWS) applications, whose performance can be continuously monitored through embedded sensors such as accelerometers and microphones to detect anomalies such as overtravel discrepancies, contamination, temperature-induced misalignment and cracks [1,2,3]. However, the integrity of these sensor data streams is often compromised by missing values arising from various factors, including sensor malfunctions, environmental disruptions, or data transmission errors [4]. Such data gaps can severely hinder the accuracy of diagnostic analyses and the effectiveness of predictive maintenance models [5,6], leading to increased downtime and operational costs. These sensor signals are characterized by high-frequency, non-stationary dynamics and subtle, transient events. This complexity makes traditional interpolation methods ineffective and requires a model capable of capturing long-range, intricate temporal dependencies.

The most elementary approaches to handling missing data in time series are statistical in nature, yet they represent the simplest and most widely implemented methods. Methods such as replacing missing values with the series mean, median, or mode are computationally trivial and easy to implement. However, they are profoundly flawed for time-series applications as they completely disregard the temporal dependencies inherent in the data and severely affect the natural variance of the signal, which can mask true anomalies and mislead downstream analyses [7]. Beyond these basic statistical measures, more sophisticated interpolation techniques have emerged as alternatives. Linear interpolation provides a straightforward approach by connecting adjacent observed values with straight lines, preserving some temporal continuity but failing to capture complex patterns. The Piecewise Cubic Hermite Interpolating Polynomial (PCHIP) [8] represents a more advanced interpolation technique that maintains smoothness while avoiding the oscillations typical of higher-order polynomial interpolation. Despite these improvements, traditional interpolation methods still struggle with longer gaps and complex temporal patterns characteristic of real-world sensor data [4,9,10,11,12,13,14]. The limitations of simple statistical methods have driven researchers toward more sophisticated Machine Learning (ML) approaches for time series imputation. K-Nearest Neighbors (KNN) imputation has emerged as one of the most effective classical techniques, consistently outperforming basic statistical methods across various datasets by leveraging similarity patterns in the data [7,15]. Random Forest and other ensemble methods have also shown promise for imputation tasks, utilizing their ability to capture non-linear relationships and feature interactions [16,17,18]. Multiple Imputation by Chained Equations represents another advanced approach that accounts for uncertainty in the imputation process by generating multiple plausible values for each missing data point [19,20]. Additionally, Expectation-Maximization algorithms and Principal Component Analysis-based methods have demonstrated effectiveness, particularly when dealing with multivariate time series where inter-variable correlations can inform the imputation process [21].

However, there exists a significant gap in the imputation of complicated signals, as traditional ML methods struggle to capture the complex temporal dependencies and non-stationary behaviors inherent in real-world sensor time series. These conventional approaches, while effective for simpler scenarios, often fail when confronted with large consecutive blocks of missing data or when dealing with multivariate time series that exhibit intricate cross-variable correlations and long-range temporal dependencies. The challenge is particularly acute in sensor applications where the data exhibits non-linear dynamics, seasonal patterns, and irregular sampling intervals. The inability of traditional methods to adequately model these complex temporal relationships can result in imputed values that significantly deviate from the true underlying signal characteristics, potentially compromising the reliability of downstream analytical tasks such as anomaly detection and predictive maintenance [5,6,22].

The emergence of deep learning methods has opened up new avenues for addressing these complex imputation challenges in time series data. Recurrent Neural Networks (RNNs) [5], Long Short-Term Memory networks (LSTM), and Gated Recurrent Units (GRU) [5] have demonstrated superior capabilities in modeling temporal dependencies and capturing long-range patterns in sequential data. These architectures excel at learning complex temporal representations through their ability to maintain hidden states and selectively forget or remember information across time steps [23]. Bidirectional approaches, such as Bidirectional LSTM, have shown particular promise by processing sequences in both forward and backward directions, enabling the model to leverage future context when imputing missing values. Advanced deep learning frameworks like Bidirectional Recurrent Imputation for Time Series (BRITS) have specifically addressed the time series imputation problem by incorporating bidirectional dynamics and treating missing values as variables in the computational graph [24]. Other deep learning approaches, such as those using Generative Adversarial Networks [25,26,27] and diffusion models [28] have also shown promise, though they can be more complex to train. Additionally, Transformer-based architectures have gained prominence due to their self-attention mechanisms, which can effectively model long-range dependencies without the sequential processing limitations of RNNs [29,30].

The Self-Attention-based Imputation for Time Series (SAITS) model represents an effective solution that addresses the fundamental limitations of previous approaches through its innovative architecture and joint optimization strategy [31]. SAITS leverages self-attention mechanisms to effectively model long-range dependencies and inter-feature correlations in multivariate time series, making it particularly suitable for complex sensor data applications. The model employs diagonally-masked self-attention (DMSA) blocks that explicitly capture both temporal dependencies and feature correlations, enabling it to handle the intricate patterns found in real-world sensor networks. The superiority of SAITS has been demonstrated across multiple domains, with studies showing it achieves the lowest mean absolute error when missing rates exceed 30%, while maintaining reasonable execution times. This effectiveness stems from its ability to simultaneously optimize imputation accuracy and preserve the temporal fidelity and spectral characteristics of the original sensor signals, making it a suitable solution for ensuring high-quality, complete sensor data in demanding industrial environments such as semiconductor manufacturing and Structural Health Monitoring (SHM) applications.Furthermore, its parallelizable attention architecture offers significant computational advantages over recurrent models, a key factor for practical deployment.

Imputation of missing values in time series underpins a wide range of applications including eye movement and gaze analytics [32,33,34], traffic flow and speed sensing [35,36,37,38,39], Internet of Things (IoT) sensor networks [40,41,42,43], weather and Photovoltaics forecasting [44,45,46], medical biosignals [25,47,48,49], smart grid energy metering [50,51,52], and financial market data [21,30], where robust gap-filling directly improves downstream tasks such as anomaly detection, forecasting, and classification.

These challenges are particularly pronounced in IoT applications, where data integrity is often compromised by transmission errors or device-level failures. In such environments, sensor networks generate multivariate time series data for tasks like forecasting and classification, but missing values can severely impact downstream analytics. Our study, while focused on semiconductor manufacturing, thus serves as a model for robust imputation in any multivariate IoT environment—such as smart cities, remote health monitoring, or environmental sensing—where high-fidelity, complete data streams are essential for maintaining accuracy and operational efficiency. The emergence of deep learning methods has opened up new avenues for addressing these complex imputation challenges in time series data. These advanced ML approaches provide a foundation for handling complex time series, which is particularly relevant to our application in probe card monitoring where sensor signals exhibit non-stationary dynamics.

The current work provides the first robust and efficient solution to a critical data integrity problem in the novel, high-stakes domain of semiconductor manufacturing. The application of the SAITS model is extended to industrial sensor data, specifically to allow for accelerometer and microphone signals that are crucial for the real-time SHM of complex PCs. While ML has been widely adopted for health monitoring, its application to PCs is a nascent but critical field of research. The complex, high-frequency signals from these sensors contain rich information about subtle failure modes, yet the successful deployment of predictive models is fundamentally predicated on the availability of complete, high-fidelity data streams. Consequently, addressing the pervasive issue of missing data is not merely a preliminary task but a novel and essential research challenge within this specific industrial domain. The SAITS framework is systematically investigated and validated as a critical preprocessing step, demonstrating its suitability for the unique demands of the EWS environment. By establishing a reliable solution for data imputation, this study provides a foundational building block for the future development of AI-assisted anomaly detection and health management systems in the semiconductor industry.

The remainder of this work is structured as follows: Section 2 details the methodology, including data preparation, missingness simulation, SAITS architecture, and evaluation metrics. Section 3 presents the experimental results, encompassing imputation performance across datasets and missing factors, as well as model training dynamics via learning curves. Section 4 discusses the findings, implications, and limitations. Finally, Section 5 concludes the study and outlines future research directions.

## 2. Methodology

This section details the methodology employed in this study. It begins with a description of the problem statement and the proposed SHM sensor network solution, followed by an in-depth explanation of the sensor data, the SAITS imputation method, and the evaluation metrics. To provide context on the PC structure, the types of failures the sensors aim to detect, and the overall performance monitoring of the PC, Figure 1 illustrates the key components and potential failure modes in one of the PCs at Technoprobe S.p.A. (Cernusco Lombardone, Lecco, Italy). The left diagram depicts the primary elements: the probe head (PH), the Printed Circuit Board (PCB), and the needles, positioned above a wafer on a chuck. The PH assembly consists of a stack of guide plates and a central housing, forming the core structure. During testing, the wafer chuck is raised to connect the needles—inserted into the PH—with the wafer pads. The right image offers a detailed view of a PC, highlighting specific failure modes, including warpage and cracks in ceramic components, loosening or damage to screws, and needle-related issues such as bending, buckling, cracking, or burning. For more comprehensive information on PC mechanics and failure detection strategies, readers are referred to previous works, such as [1,2,3].

The effective real-time monitoring and management of PCs in EWS applications are paramount to maintaining high efficiency and reliability in semiconductor manufacturing. This section details the sensor network design for detecting critical PC failure modes, including the challenges associated with these failures and the specific sensors employed, with a focus on accelerometer and microphone sensors for imputation.

### 2.1. Probe Card Failure Modes and Associated Challenges

A historical failure data analysis has identified several critical failure modes in PCs, which can be broadly categorized into needle-related, mechanical, and electrical failures. The current research has prioritized these issues based on their occurrence frequency and repair time. The key failure modes and their associated challenges include the following:Temperature Differences between Wafer and PCDescription: Temperature variations across different components of the PC and the wafer can lead to misalignment.Challenges: They represent a possible root cause of open/short-circuit problems, which are the most frequent electrical failure modes, and can arise from insufficient soaking time.Sensor for Detection: Multiple temperature sensors are embedded at strategic locations within the PC for real-time monitoring of thermal variations.Cracks in PC PlatesDescription: Critical mechanical failures show up where cracks originate at stress concentration points, like probe holes and screw holes. These are sometimes difficult to detect by optical inspection and can occur in internal plates, not externally visible.Challenges: They demand early detection to prevent catastrophic failure and ensure wafer testing integrity.Sensor for Detection: Finite Element simulations show that cracks induce discernible changes in acceleration frequency spectra. Microphones are therefore integrated into the sensor network for real-time monitoring of Frequency Response Functions (FRFs), to capture these frequency anomalies.Loosening of Screws in PH AssembliesDescription: PH assemblies are secured by screws, and their integrity is vital for mechanical stability and alignment. Improperly tightened or loosened screws, due to vibrations or thermal cycling, can lead to tip misalignment.Challenges: Tip misalignment is a significant contributor to electrical failure modes, due to inconsistent wafer pad contact.Sensor for Detection: Similarly to cracks, loosened screws affect the vibrational characteristics and FRFs of the PH assembly. Microphones are employed to detect this failure mode through frequency analysis and comparison with baseline profiles.Contamination of Probe TipsDescription: Accumulation of contaminants (e.g., aluminum oxide, environmental particles) on probe tips.Challenges: It causes open circuits, unintended shorts, increased CRES, and false identification of functional devices as defective, leading to reduced yield and higher costs. Online cleaning processes, while necessary, consume probe tips and often do not restore optimal conditions.Sensor for Detection: Accelerometers are used for the automatic estimation of touchdown counts and PC lifespan, providing mechanical data on usage and wear. These data, combined with electrical test data from the automatic test equipment, help predict the optimal cleaning time.

### 2.2. Sensor Network and Data Imputation Focus

To address these challenges, a comprehensive sensor network system has been developed and integrated into operational PCs. This network comprises:Accelerometers, for tracking probe touchdowns and predicting maintenance needs.Microphones, for detecting structural issues like cracks or loose components through acoustic monitoring.Temperature Sensors, to identify misalignment due to thermal expansion.

The sensor network facilitates real-time collection of data, which are then stored to develop ML models for anomaly detection, failure forecasting through predictive maintenance, and proactive management [3]. Figure 2 shows the proposed sensor system used for real-time online monitoring and management of the PC.

This investigation focuses on the imputation of multivariate time series from an analog microphone (MIC) and two accelerometers (ACC1 and ACC2), which are frequently corrupted by missing data that hinders subsequent analysis. These specific sensor signals are characterized by high-frequency, non-stationary dynamics, which are essential for detecting failures like cracks or contamination. The primary hypothesis of this study is that advanced imputation models are required to preserve the complex dynamics, a task where traditional methods fail. Signals from temperature sensors were excluded, as our preliminary analysis showed their low temporal variance does not require advanced imputation techniques. The proposed methodology employs a deep learning approach, the SAITS model, to reconstruct the incomplete signals. This imputation constitutes a critical preprocessing step for downstream ML applications, such as classification and anomaly detection, which require complete data streams. Ultimately, achieving accurate signal reconstruction is a prerequisite for enabling robust, AI-assisted systems for the real-time monitoring and performance optimization of the PCs. The framework implements a multi-step pipeline encompassing data preprocessing, imputation, and rigorous evaluation of the reconstructed signal quality.

### 2.3. Data Acquisition and Experimental Setup

The time-series data for this study were acquired from a custom sensor network during the real-time operation of PCs in EWS applications. This network, depicted in Figure 2, integrates the different sensors managed by a dedicated microcontroller (MCU). For this work, the focus is on the data streams from two 3-axis accelerometers and one analog microphone. The MCU was operated in an offline mode to collect a comprehensive dataset for the subsequent training and validation of the imputation models, which are ultimately intended for deployment in real-time SHM.

The acquired raw time-series signals sampled at 1 kHz were first preprocessed by segmenting them into fixed-length windows of 1000 time steps. A key methodological contribution of this work is this signal segmentation strategy, which is intentionally designed as a prerequisite for a deployable online system. By partitioning the data into discrete one-second windows made of 1000 time steps, the framework is designed to be capable of processing incoming sensor data in a sequential, near real-time manner. While the model in this study is trained offline, this windowing approach is the foundational step for translating the imputation model into a practical tool for the continuous, online SHM of PCs in a live manufacturing environment. This segmentation process yielded a dataset of 3D arrays with dimensions of (9401, 1000, 3) for the accelerometers and (9131, 1000, 1) for the microphone.

To create a controlled experimental setup for evaluating the imputation performance, artificial data gaps were programmatically introduced into the complete windows, thereby establishing a ground truth. The mechanisms underlying missing data can be categorized into three distinct types, according to how the absence of data relates to the dataset characteristics [53,54]: (i) Missing Completely at Random (MCAR), which describes scenarios where data absence has no systematic relationship with any observed or unobserved variables; (ii) Missing at Random (MAR), which occurs when the probability of data absence depends on other observable variables within the dataset, but not on the missing values themselves; and (iii) Missing Not at Random (MNAR), which arises when the likelihood of missingness is influenced by the unobserved values themselves or by factors not captured in the dataset [36,55,56,57]. In our approach, the missingness is generated in contiguous blocks to simulate realistic scenarios such as intermittent sensor failure. These rates were selected to represent both moderate (10%) and severe (30%) data loss scenarios. Specifically, shorter blocks of 5 time steps for the 10% missing factor were used to model brief dropouts, and much longer blocks of 20 time steps for the 30% missing factor to simulate more severe, prolonged sensor outages. This increase in block length at the higher missing rate is a deliberate design choice to create a more challenging benchmark, testing the models ability to reconstruct signals across extended data gaps where local context is entirely absent. In this study, the missing data simulation employs a strictly MCAR mechanism, rather than MNAR, to generate contiguous blocks of missing values in the sensor signals. Specifically, the starting positions of these missing blocks are selected randomly, ensuring complete independence from the underlying signal values. This MCAR approach establishes a reproducible and unbiased benchmark, as the missing data pattern can be precisely replicated using a fixed random seed. By decoupling the missingness from the data values, it creates a fair evaluation framework for assessing each model ability to reconstruct intricate temporal patterns solely from the adjacent observed data.

Prior to model training, each feature was normalized using the mean and standard deviation of the training set to ensure all variables contributed equally during model optimization. The dataset was then partitioned into training (80%), validation (10%), and testing (10%) subsets to facilitate an unbiased evaluation. A corresponding denormalization function was implemented to revert imputed values to their original scale for error analysis.

### 2.4. SAITS Model Architecture and Configuration

We adopted the SAITS model [31] for missing value imputation, leveraging its Transformer-inspired architecture to capture both temporal and cross-feature dependencies in multivariate signals. SAITS is particularly suited for time series imputation due to its DMSA mechanism, which enforces causality and focuses attention on observed values while iteratively refining imputations.

The core architecture of SAITS, as illustrated in Figure 3, consists of two parallel DMSA blocks that process the input data—a concatenation of feature vectors and binary missing masks—after linear projection and positional encoding. Each DMSA block comprises a stack of *L* layers, where each layer includes a diagonal-masked multi-head attention (MHA) module, followed by add & norm operations (i.e., a residual connection that adds the sub-layer input to its output, combined with layer normalization to stabilize gradients and improve training), a feed-forward network (FFN), and another add & norm. The outputs from the two DMSA blocks are combined via a weighted mechanism to produce a refined representation. This refined representation is then projected back to the original feature space through a linear transformation (typically a fully connected layer) to yield the final imputed values, with a replacement step integrating preliminary imputations back into the process for iterative refinement.

In the implementation, the SAITS hyperparameters were optimized through a grid search over the number of layers per DMSA block (*L*: 2 or 3), embedding dimension ($d_{\mathrm{model}}$: 64, 128, or 256), and learning rate (0.001, 0.0005, or 0.0001), with fixed epochs of 300 and early stopping (patience = 50) based on the validation loss. The search evaluated 18 combinations per dataset and missing rate, selecting the configuration with the lowest validation MSE on imputed missing positions. Detailed results are provided in Appendix A, where it is shown that deeper models (L=3) and larger embeddings ($d_{\mathrm{model}}=256$) consistently performed the best. For example, optimal settings for ACC1 at 10% missing were L=3, $d_{\mathrm{model}}=256$, and learning rate of 0.0005. Key and value dimensions were scaled dynamically as $d_{k}=d_{v}=d_{\mathrm{model}}/4$ (e.g., 64 for $d_{\mathrm{model}}=256$), with 4 attention heads ($n_{\mathrm{heads}}=4$). The FFN hidden dimension ($d_{\mathrm{ffn}}$) was set to twice $d_{\mathrm{model}}$ for balanced capacity. A dropout rate of 0.05 was also applied for regularization. Data normalization was performed per sample using observed values only. The final model was retrained on combined train and validation data using the optimal hyperparameters, employing the Adam optimizer [58] with weight decay of $1\times 10^{-5}$. Training spanned up to 300 epochs with a batch size of 64. The implementation utilized the PyPOTS library [59], with computations on an available CUDA-enabled GPU.

### 2.5. Comparison Methods

To rigorously evaluate the performance of the SAITS model, we compare it against a set of established traditional techniques and a state-of-the-art deep learning model. These comparison methods were selected to represent a range of complexities, from simple interpolation to more sophisticated recurrent architectures. Comparison Methods were chosen to represent key categories: Linear (simple), PCHIP (interpolation), KNN (ML), and BRITS (DL). More recent Transformers like Informer were not included as the focus is on imputation efficiency; SAITS single-pass inference outperforms iterative alternatives like diffusion models in speed. In contrast, diffusion models are generative and operate through an iterative denoising process. To perform imputation, they must execute a reverse diffusion chain, which involves multiple (often hundreds or thousands of) sequential steps of a neural network to transform noise into a coherent signal.

Linear Interpolation: This is one of the most straightforward imputation techniques. It treats missing values as points on a straight line connecting the last observed data point before the gap, and the first observed data point after the gap itself. While computationally efficient, it assumes a linear trend within the missing interval and fails to capture any non-linear dynamics.Piecewise Cubic Hermite Interpolating Polynomial (PCHIP): PCHIP is a more advanced interpolation method that fits a cubic polynomial to the data segments between observed points. Unlike standard cubic splines that can overshoot and introduce artificial oscillations, PCHIP is shape-preserving, ensuring that the imputed segment maintains the monotonicity of the surrounding data.K-Nearest Neighbors (KNN): KNN is a classical ML algorithm that imputes missing values based on similarity. For a given sample with missing data, it identifies the *k* most similar complete samples (neighbors) in the training dataset, typically using the Euclidean distance. The missing values are then filled using a weighted average of the values from these neighbors at the corresponding time steps. In our implementation, we use k=5, a common default value for this method, and weight the contributions of neighbors by the inverse of their distance.Bidirectional Recurrent Imputation for Time Series (BRITS): BRITS serves as a key comparison method in our evaluation, as it represents a state-of-the-art deep learning approach for multivariate time series imputation. Introduced in [24], this method employs bidirectional RNNs, typically implemented with GRUs or LSTM cells, to capture temporal dependencies in both forward and backward directions. By modeling the data inherent correlations across features and time steps, BRITS directly learns to estimate missing values through a unified architecture that integrates imputation and classification tasks, enhancing robustness without auxiliary models. In the present implementation, BRITS was adapted to handle the sensor datasets through configuration of bidirectional RNN layers for processing multivariate inputs, with optimization based on mean squared error loss to ensure imputation accuracy. This method was selected for its demonstrated efficacy on irregular time series, offering a rigorous benchmark to evaluate advancements of SAITS in self-attention mechanisms that enable superior modeling of long-range dependencies and spectral preservation—i.e., the maintenance of frequency-domain characteristics of the signal, such as power spectrum and harmonic content—in industrial monitoring contexts.

### 2.6. Evaluation Metrics

To provide a comprehensive assessment of the imputation quality, a set of time-domain and frequency-domain metrics is computed, comparing the imputed signals against the original (ground-truth) signals at the missing locations. These metrics quantify different aspects of imputation accuracy: time-domain metrics measure point-wise errors and correlations in the temporal sequence, while frequency-domain metrics evaluate the preservation of spectral properties, which is crucial for signals like vibrations where frequency content informs physical phenomena such as resonances or anomalies.

Time-domain evaluation focuses on the sample-wise deviation between the ground truth and the imputed signals. The subsequent metrics are calculated exclusively on the indices where data points were artificially removed, as summarized in Table 1. MAE represents the average absolute difference, capturing the overall magnitude of errors; MRE normalizes errors relative to ground-truth values, which is useful for scale-invariant assessment; RMSE emphasizes larger errors through squaring, providing an indication of typical error magnitude; Sim quantifies structural similarity via correlation, indicating how well patterns are preserved; and FSD normalizes RMSE by the standard deviation of the signal, offering a relative measure of error variability. For these metrics, lower values of MAE, MRE, RMSE, and FSD signify superior imputation performance, while a Sim value approaching unity indicates strong correlation with the ground truth. In Table 1, $N_{\mathrm{imp}}$ denotes the total number of imputed samples, I represents the set of imputed indices (positions where data was missing), and $\bar{s}$, $\bar{\hat{s}}$ are the mean values of the ground-truth signal (*s*) and the imputed signal ($\hat{s}$), respectively, calculated only over I.

Spectral-domain metrics are employed to assess how well the imputation process preserves the frequency characteristics of the time series. Unlike time-domain metrics, these are computed on the full signal length, as Fourier analysis necessitates a complete sequence to generate a valid spectrum, as detailed in Table 2. ${\mathrm{RMSE}}_{F}$ measures overall spectral error across all frequencies; ${\mathrm{RMSE}}_{F_{\mathrm{Low}}}$ focuses on low-frequency components (0–15 Hz), which often represent baseline trends or slow-varying dynamics in sensor data; and ${\mathrm{RMSE}}_{F_{\mathrm{High}}}$ evaluates high-frequency bands (>150 Hz), capturing noise or rapid oscillations that could indicate faults. For frequency-domain metrics, smaller RMSE values across the full, low, and high-frequency bands demonstrate more accurate reconstruction of the spectral integrity of the signal. In Table 2, $S_{k}$ and ${\hat{S}}_{k}$ are the normalized coefficients from the one-sided Discrete Fourier Transform of the original signal *s* and the imputed signal $\hat{s}$, respectively, and *K* is the number of frequency bins. Metrics are computed on the full signal length.

An evaluation function iterates through each feature of the sensor data, applies the defined metrics, and stores the results.

## 3. Results

In this section, the imputation performance is assessed for three sensor signals: ACC1 and ACC2 (multivariate accelerometer data with three features) and MIC (univariate microphone signal data). Results are reported for missing factors of 0.1 and 0.3, focusing on SAITS and comparison methods (BRITS, PCHIP, Linear, KNN). The metrics are computed on the test set.

### 3.1. ACC1 Signals Imputation at 10% Missing Rate

Table 3 summarizes the imputation performance for the ACC1 dataset with a 10% missing data rate. The results show lower errors for SAITS model over all comparison methods across the three features. In the time domain, SAITS attained the lowest MAE values of 4.31, 3.45, and 13.31 for features 0, 1, and 2, respectively, coupled with robust Sim scores ranging from 0.69 to 0.96, which collectively signify exceptional correlation with ground truth signals and high-fidelity reconstruction.

In contrast, conventional methods such as PCHIP and Linear Interpolation generated considerably larger errors; for example, the MAE obtained by PCHIP for Feature 0 reached 23.31, exceeding that of SAITS by a factor of five. Notably, these simpler techniques, including KNN, frequently yielded negative Sim values (e.g., −0.15 for PCHIP in Feature 0), implying an inverse correlation with the true signal. This occurs when interpolation methods draw a simple line across a complex dynamics (e.g., a peak or trough), resulting in an imputed segment that trends in the opposite direction of the actual data, underscoring their inability to capture temporal dynamics. Moreover, SAITS distinguished itself in preserving spectral characteristics, registering the lowest RMSE in the frequency domain (${\mathrm{RMSE}}_{F}$) across features (e.g., 38.16 for Feature 0), while comparison methods induced severe distortions, with the ${\mathrm{RMSE}}_{F}$ obtained by PCHIP escalating to 906.70 for Feature 2. Although BRITS emerged as a strong contender, SAITS maintained consistent efficacy in both temporal accuracy and frequency-domain preservation, affirming its suitability for challenging imputation scenarios.

### 3.2. ACC1 Signals Imputation at 30% Missing Rate

As the data loss escalates to a 30% missing factor, Table 4 illustrates the robustness of the SAITS model in a more challenging imputation scenario. Although all methods encountered performance degradation, SAITS preserved its superior accuracy, with RMSE increases of approximately 1.1–2× relative to the 10% missing rate scenario across features. In contrast, comparison methods such as PCHIP exhibited error escalations of 4–7×, underscoring their vulnerability to extended gaps. Critically, SAITS maintained a strong positive correlation with the ground truth, achieving Sim scores of 0.89, 0.59, and 0.87 for features 0, 1, and 2, respectively, which reflect its ability to retain signal fidelity even under substantial data loss.

This resilience stands in sharp contrast to the pronounced collapse of traditional methods. PCHIP and Linear Interpolation not only produced dramatically higher time-domain errors—for instance, the MAE obtained by PCHIP for Feature 2 reached 191.88, over eight times that of SAITS at 23.01—but also yielded Sim scores near zero or negative (e.g., −0.05 for PCHIP across features), indicating a fundamental failure to reconstruct underlying signal dynamics amid longer contiguous gaps. The disparity was even more evident in the frequency domain, where comparison methods induced extreme spectral distortions; for example, the ${\mathrm{RMSE}}_{F}$ obtained by PCHIP for Feature 2 surpassed 13,498, whereas SAITS contained the error at 367.51. While BRITS performed competitively, often rivaling SAITS in select time-domain metrics (e.g., lower MRE for Features 0), SAITS again demonstrated a consistent advantage in balancing overall temporal and spectral preservation, particularly in high-missing-rate contexts. These findings affirm the advanced capacity of SAITS to manage extensive, block-wise missing data, positioning it as a highly reliable solution for real-world sensor imputation in industrial monitoring applications.

### 3.3. ACC2 Signals Imputation at 10% Missing Rate

For the ACC2 dataset at a 10% missing factor, Table 5 reaffirms the trends observed in ACC1, further solidifying the SAITS model superior imputation performance across the diverse sensor datasets. SAITS consistently delivered the lowest errors regarding all the metrics, enabling highly accurate signal reconstruction, as evidenced by its near-perfect Sim scores of 0.96, 0.74, and 0.98 for features 0, 1, and 2, respectively, alongside low MAE values ranging from 4.81 to 14.72.

The performance disparity between SAITS and comparison methods remained substantial. For instance, for Feature 2, the MAE of 109.63 obtained by PCHIP was over seven times higher than the value of 14.72 achieved by SAITS, while Linear Interpolation and KNN fared similarly poorly. Moreover, these traditional approaches struggled to preserve temporal structure, often producing negative Sim scores—such as the value of −0.45 obtained by KNN for Feature 0—which signify an anticorrelation with the ground truth and highlight their inadequacy for complex, multivariate time-series data. In the frequency domain, SAITS excelled at maintaining spectral integrity, with the lowest${\mathrm{RMSE}}_{F}$ values (e.g., 130.33 for Feature 2), whereas comparison methods induced severe distortions; the ${\mathrm{RMSE}}_{F}$ obtained by PCHIP for the same feature was nearly an order of magnitude greater, reaching 1308. Although BRITS showed competitive results as the runner-up, SAITS maintained a clear advantage in holistic fidelity across both domains. These outcomes underscore the robust and consistent capabilities of SAITS for imputing similar multivariate sensor streams in real-world applications.

### 3.4. ACC2 Signals Imputation at 30% Missing Rate

Table 6 shows the results for ACC2 at a 0.3 missing factor, revealing how imputation performance degrades under higher missingness while maintaining the established performance hierarchy. SAITS continued to demonstrate superior imputation capabilities across all features, though the increased missing data introduced greater challenges for all methods. SAITS achieved strong Sim scores of 0.92, 0.67, and 0.92 for features 0, 1, and 2, respectively, with MAE values ranging from 6.45 to 27.61, representing the most accurate reconstructions among all competing approaches.

The performance gap between SAITS and traditional methods widened considerably at this higher missing rate. For Feature 0, the MAE obtained by PCHIP (41.94) was approximately six times higher than that obtained by SAITS (6.97), while the RMSE of 135.07 was nearly fifteen times larger. Similarly, for Feature 2, the MAE obtained by PCHIP reached 150.36 compared to the value of 27.61 achieved by SAITS, demonstrating the severe degradation of interpolation-based methods under substantial missingness. Traditional approaches continued to struggle with temporal coherence, though some improvement was observed; for instance, KNN achieved a slightly positive Sim score of 0.04 for Feature 0, suggesting marginal correlation recovery, whereas PCHIP and Linear Interpolation remained near zero or negative. In the frequency domain, the disparity became even more pronounced. SAITS maintained relatively controlled ${\mathrm{RMSE}}_{F}$ values (e.g., 109.75 for Feature 0 and 438.42 for Feature 2), while comparison methods exhibited catastrophic spectral distortions. The ${\mathrm{RMSE}}_{F}$ obtained by PCHIP for Feature 0 reached 2233.13—over twenty times higher than that of SAITS—and for Feature 2, it soared to 6596.02, indicating severe frequency domain corruption. The low-frequency components were particularly affected, with the ${\mathrm{RMSE}}_{F_{\mathrm{Low}}}$ values obtained by PCHIP reaching 6567.73 and 19,606.04 for Features 0 and 2, respectively. BRITS again emerged as the closest competitor to SAITS, showing competitive performance across most metrics, though SAITS retained a consistent edge in both time and frequency domain accuracy. These findings demonstrate that the deep learning architecture of SAITS maintains robust performance even under challenging conditions with substantial missing data, making it particularly valuable for real-world sensor applications where data loss rates can be unpredictable and severe.

### 3.5. MIC Sensor Imputation at 10% Missing Rate

Table 7 summarizes the imputation performance regarding the MIC sensor, which comprises a single feature, at a 10% missing data rate. The results highlight the SAITS model superiority over the comparison methods, achieving the lowest MAE of 712.78 and a high Sim score of 0.80, which are indicative of exceptional temporal fidelity and close alignment with the ground truth signal. In contrast, conventional approaches such as PCHIP and Linear Interpolation produced markedly larger errors, with PCHIP MAE reaching 2028.70—nearly three times that of SAITS—while its Sim score is only 0.09, signifying a fundamental shortcoming in reconstructing the signal dynamics. KNN similarly underperformed, yielding a Sim of 0.00, indicating no correlation with the original signal. The frequency-domain analysis further accentuated these disparities, where SAITS preserved spectral integrity with the lowest${\mathrm{RMSE}}_{F}$ of 7920.46, while comparison methods induced significant distortions; for instance, PCHIP ${\mathrm{RMSE}}_{F}$ escalated to 22,956.49. Although BRITS emerged as the strongest comparison method with competitive metrics (e.g., MAE of 1001.92 and Sim of 0.63), SAITS maintained a clear advantage in both time- and frequency-domain preservation. These findings reinforce SAITS efficacy for univariate time-series imputation in sensor-based applications, ensuring reliable data integrity under moderate missingness.

### 3.6. MIC Sensor Imputation at 30% Missing Rate

Finally, Table 8 summarizes the imputation performance for the MIC sensor data, featuring again a single channel, under a 30% missing data rate. The outcomes underscore the SAITS model resilience in handling elevated missingness. SAITS again achieves the best performance, with the lowest MAE of 976.14 and a robust Sim score of 0.73, reflecting strong temporal alignment and effective reconstruction despite the increased challenge. Conversely, traditional methods like PCHIP and Linear Interpolation incurred substantially amplified errors. This is exemplified by the MAE of 3712.99 obtained by PCHIP—over 3.8 times the value achieved by SAITS—and a near-zero Sim score of 0.05, revealing their inability to model signal dynamics amid extended gaps. KNN likewise faltered, with a Sim of 0.05. The frequency-domain disparities are particularly stark, as SAITS minimizes spectral distortions with an ${\mathrm{RMSE}}_{F}$ of 17,747.03, whereas comparison methods generate severe artifacts; PCHIP ${\mathrm{RMSE}}_{F}$, for instance, soars to 195,058.93, an order of magnitude worse, which would compromise any downstream analysis. While BRITS proves a capable alternative (MAE of 1336.50 and Sim of 0.50), SAITS retains a definitive lead in integrated time- and frequency-domain fidelity. These results validate once more the SAITS proficiency for univariate sensor data imputation in high-missing-rate environments, enhancing reliability for industrial diagnostics.

### 3.7. Model Training Dynamics

To provide deeper insights into the training process of the SAITS model, the learning curves were analyzed for each dataset and missing factor, as depicted in Figure 4 and Figure 5. These graphs illustrate the evolution of the training loss (MAE, computed on the imputed values relative to the ground truth) and validation MSE over 300 epochs for accelerometer datasets (ACC1 and ACC2) and up to 500 epochs for the microphone dataset (MIC). MAE was used for the training loss due to its robustness to outliers, while MSE was used for validation to heavily penalize large errors during hyperparameter selection. The training MAE decreases rapidly in the initial epochs (typically within the first 50), indicating an efficient capture of the underlying patterns in the time series. This is followed by a gradual plateau, suggesting convergence without signs of overfitting, as the validation MSE closely mirrors the training trajectory without upward divergence. For lower missing factors (0.1), the curves exhibit smoother and faster stabilization, with final validation MSE values around 0.2–0.25 for ACC datasets and 0.3 for MIC, to reflect the model proficiency in handling short gaps. In contrast, at higher missing factors (i.e., 0.3), the validation MSE shows slightly more oscillations early on, stabilizing at higher levels (0.4–0.7), which aligns with the increased challenge of imputing longer blocks. Notably, the univariate MIC dataset displays greater volatility in validation loss for the 0.3 case, underscoring the benefits of multivariate contexts in ACC datasets where cross-feature dependencies aid imputation.

Overall, these dynamics confirm the model robustness, with early stopping mechanisms (patience = 50) effectively preventing unnecessary epochs while ensuring generalizable imputations.

### 3.8. Statistical Analysis

To statistically validate the observed performance differences, Wilcoxon signed-rank tests were conducted on the per-sample error values for each metric and feature across the entire test set. The detailed p-values for the hundreds of resulting comparisons are not presented in a table for the sake of brevity, as they were overwhelmingly consistent. The results confirm that SAITS significantly outperforms all traditional baseline methods (PCHIP, Linear, and KNN) in all evaluated scenarios (all features, all datasets, all metrics) with p<0.001. When compared to the deep learning baseline (BRITS), SAITS also demonstrated a statistically significant advantage (p<0.05) in the vast majority of cases across all datasets and missing rates. A non-significant difference was observed in only one specific instance: the $RMSE_{F_{\mathrm{Low}}}$ metric for Feature 1 of the ACC2 dataset at a 30% missing rate (p=0.270). These statistical findings strongly support the quantitative results, confirming that SAITS provides a consistently and significantly more accurate imputation than traditional methods and a highly competitive, often superior, performance compared to BRITS.

### 3.9. Qualitative Visual Analysis

Figure 6 provides an exemplary qualitative visualization of the performance of the different imputation methods on a representative segment of the ACC2 test dataset. The plots clearly illustrate the superior performance of the deep learning models, SAITS (red) and BRITS (purple), which closely track the ground truth signal (dashed grey) within the missing intervals. Both models successfully reconstruct complex dynamics, accurately capturing the amplitude and shape of the underlying signal, such as the sharp peaks and troughs around time steps 512 and 526.

In contrast, the traditional methods fail to capture these essential characteristics. Linear interpolation (green) and PCHIP (blue) merely bridge the gap endpoints with overly simplistic lines, completely missing the intra-gap variations. KNN (orange) produces erratic and inaccurate reconstructions, demonstrating its inability to find appropriate neighboring patterns for these complex signals. This visual evidence reinforces the quantitative results, highlighting how SAITS and BRITS preserve the true signal fidelity, whereas comparison methods introduce a significant distortion.

## 4. Discussion

The findings of this study consistently validate SAITS as a robust and effective imputation framework for the high-frequency, complex sensor data characterizing to probe card monitoring. A primary conclusion, supported by rigorous statistical analysis, is the stark performance gap between SAITS and traditional methods. The results demonstrate that while SAITS successfully preserves critical temporal and spectral signal characteristics—even under high data loss (30%)—traditional interpolation methods like PCHIP and Linear fail completely. This failure, often manifesting as severe signal distortion and inverse correlations (negative Sim scores), underscores their unsuitability for all but the most basic signals. This resilience to extended data gaps highlights the necessity of advanced models for ensuring data integrity. Furthermore, the evaluation provides a nuanced comparison to the BRITS deep learning baseline, revealing a critical trade-off between performance, consistency, and computational efficiency, which is discussed in the following paragraphs.

The performance gap between SAITS and traditional imputation techniques becomes particularly stark under challenging conditions with high missing rates. The quantitative results reveal a dramatic and consistent advantage for SAITS. Based on a comprehensive analysis of all 14 experimental scenarios (all features across all datasets and missing rates) against the three traditional methods (PCHIP, Linear, and KNN), SAITS achieved an average relative reduction in MAE of 66.6%. This improvement was robust across all conditions, with the relative MAE reduction ranging from 24.8% to a maximum of 88.0%. This substantial improvement underscores a fundamental limitation of traditional approaches: methods like linear and PCHIP interpolation are structurally incapable of recreating complex, non-linear signal dynamics, as they merely bridge the endpoints of a data gap with simplistic assumptions. In contrast, the SAITS model leverages its self-attention mechanism to learn intricate temporal dependencies and cross-feature correlations from the surrounding observed data, enabling it to reconstruct the underlying patterns within the missing segments with high fidelity. This capability is not just a matter of numerical accuracy but is critical for preserving the physical meaning of the sensor signals—such as the high-frequency, non-stationary vibrations from accelerometers and acoustic transients from microphones—preventing the introduction of artifacts that could severely compromise the reliability of downstream anomaly detection and health monitoring systems in semiconductor probe card applications.

When compared to BRITS, another deep learning comparison method, SAITS delivers highly competitive results, often edging ahead in overall metrics. However, a balanced analysis reveals that BRITS occasionally surpasses SAITS in specific cases, such as lower MAE and RMSE for certain features in the ACC datasets at 30% missing rates (e.g., for ACC2 Feature 1, BRITS achieves an MAE of 6.36 compared to 6.45 for SAITS, along with a superior RMSE of 7.97 compared to 8.10, as shown in Table 6). Similarly, for ACC2 Feature 2 at 30% missing rate (Table 6), BRITS obtains a lower MAE of 23.75 versus 27.61 for SAITS, along with a superior RMSE of 30.69 compared to 37.01. This variability may stem from the bidirectional recurrent architecture of BRITS, which excels at capturing sequential dependencies in scenarios with pronounced temporal correlations, albeit at the cost of handling longer-range patterns less efficiently than the self-attention mechanism of SAITS. Notably, despite these isolated instances where BRITS demonstrates marginal superiority in point-wise accuracy for specific features, SAITS maintains a more consistent advantage across the majority of metrics and features. For instance, in the ACC1 30% missing scenario, SAITS achieves better performance for Features 0 and 1 (e.g., MAE of 6.84 vs. 7.21 for Feature 0), and demonstrates substantially superior frequency-domain preservation across all features. For the univariate MIC dataset, SAITS maintains a consistent advantage across both missing rates, suggesting that its DMSA blocks are particularly apt at leveraging intra-signal dependencies without relying on multivariate cues.

A critical trade-off emerges when considering computational efficiency, as revealed by the training speed analysis conducted on an NVIDIA GeForce RTX 3090 card. SAITS completes a single training epoch in approximately 49 s, over 20 times faster than BRITS, which requires 19–20 min per epoch. Consequently, SAITS can run over 300 epochs in under 4 h, while BRITS manages only 77 epochs in more than 25 h under identical conditions. This speed advantage likely arises from the SAITS parallelizable attention-based architecture, which processes sequences holistically without the sequential bottlenecks inherent in BRITS recurrent structure. This efficiency extends to inference, where SAITS processed imputations in 0.2571 s versus 2.1601 s for BRITS (an 8.4× speedup), a critical factor for real-time deployment. In practical terms, this efficiency enables rapid experimentation, hyperparameter tuning, and frequent model retraining—essential for industrial settings where sensor data streams evolve dynamically and computational resources may be constrained. While BRITS occasional metric superiority highlights its strengths in precision for select signals, SAITS offers a more pragmatic alternative, balancing competitive performance with feasibility for time-sensitive applications.

The learning curves further support the claim related to SAITS robustness, showing rapid initial convergence and stable validation MSE without overfitting, even at higher missing rates. For multivariate ACC datasets, cross-feature correlations appear to enhance imputation stability, whereas the univariate MIC exhibits more volatility, suggesting opportunities for domain-specific adaptations. Overall, these findings position SAITS as a versatile tool for sensor data imputation, with implications extending beyond PC monitoring to broader IoT and SHM domains.

While this study demonstrates SAITS efficacy on simulated missingness from real sensor data, several limitations must be acknowledged. The primary limitation is the lack of real-time implementation testing; future work will prioritize this online deployment in live manufacturing settings and evaluate performance across broader domains and more diverse datasets, including varying sensor types and environmental conditions. Second, our results revealed a potential sensitivity to low-amplitude signal features (evident in the elevated MRE scores for some features) which warrants further investigation. Third, this study used an MCAR simulation, and further validation is needed under more complex, adaptive missingness patterns (e.g., MNAR). Finally, while the current study focuses on accelerometers and microphones—which were specifically chosen based on historical failure data, as they are essential for detecting the most frequently occurring mechanical, electrical, and needle-related failure modes—the SAITS imputation method is generalizable. Thus, future work will also evaluate its performance on signals from other sensors in our network (e.g., deformation sensors) which are intended for failure modes that, while still relevant, occur less frequently. A further enhancement will involve integrating explainability features, such as attention maps, to interpret imputation decisions and strengthen the model applicability in complex, safety-critical sensor networks.

## 5. Conclusions

This study has demonstrated SAITS as an efficient and effective deep learning framework for imputing missing values in sensor time series, tailored for real-time PC monitoring in semiconductor manufacturing. By leveraging self-attention mechanisms, SAITS consistently outperforms traditional comparison methods and provides competitive results against BRITS, while offering a significant computational advantage—over 20 times faster training per epoch. This speed–efficiency trade-off, without substantial sacrifices to imputation quality, makes SAITS particularly suitable for resource-constrained industrial environments, ensuring reliable data reconstruction for anomaly detection and predictive maintenance. Ultimately, this work advances resilient AI-driven monitoring, contributing to reduced downtime and improved operational efficiency in high-stakes manufacturing processes.

The dataset utilized in this study consists of signals from both defect-free (baseline) and defective probe cards, enabling an evaluation of imputation performance across real-world variability, including failure modes. This inclusion provides preliminary insights into the ability of SAITS to reconstruct signals affected by defects, though a more comprehensive evaluation of its effectiveness in such scenarios is warranted.

In the discussion of computational efficiency, SAITS not only excels in training speed but also demonstrates superior inference computation time. This inference efficiency is particularly critical for real-time applications in semiconductor manufacturing, where rapid data reconstruction is essential to minimize system latency and enable timely process monitoring and decision-making. The reduced computational overhead positions SAITS as a viable solution for deployment in time-sensitive industrial environments where millisecond-level response times directly impact production quality and operational efficiency.

Future research directions include optimizing SAITS to further narrow performance gaps in scenarios where BRITS excels, such as through hybrid attention-recurrent modules, while preserving its core efficiency. More specifically, future work will focus on real-time implementation, exploring optimization techniques to integrate the SAITS framework into operational environments and deploy it on edge devices. We will also investigate adaptive missingness, evaluating the model under more complex and dynamically changing data patterns. Other avenues include sensor fusion, by extending the framework to explicitly model relationships between different sensor types, and integration with predictive models. This final step will involve building dedicated models that leverage SAITS-imputed signals to enhance accuracy in detecting anomalies and forecasting failures, thereby quantifying the end-to-end impact of improved data quality. This research aimed to contribute to the development of more resilient and intelligent industrial monitoring systems, ultimately leading to improved reliability, reduced downtime, and enhanced efficiency in manufacturing processes.

## Figures and Tables

**Figure 1 sensors-25-07194-f001:**
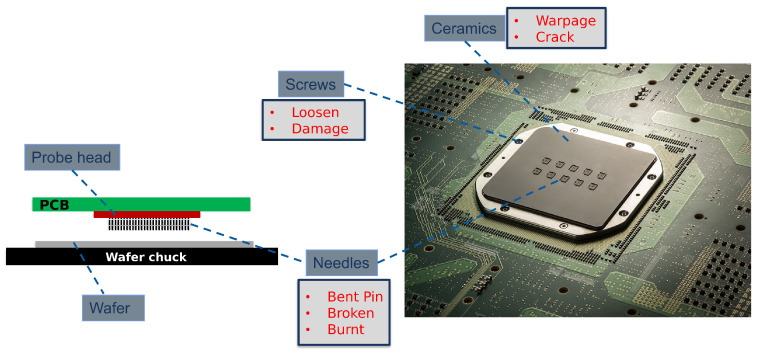
Structure of a probe card; a sketch (**left**) and highlighted failure modes (**right**) [3].

**Figure 2 sensors-25-07194-f002:**
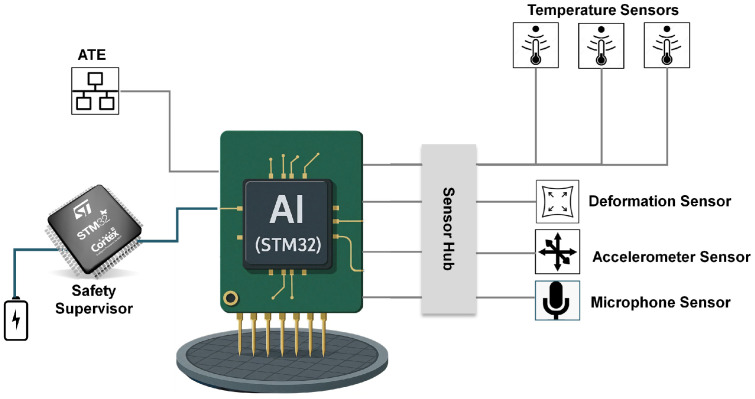
Proposed sensor system for SHM of the Probe Card with accelerometers and microphone indicated [3].

**Figure 3 sensors-25-07194-f003:**
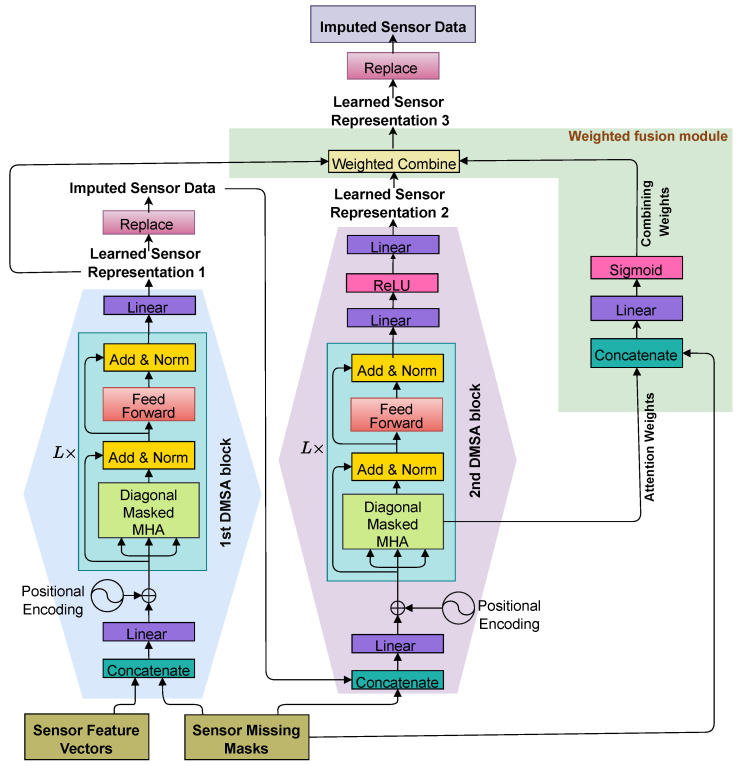
Architecture of the SAITS model, adapted from [31] by modified it to reflect the “Sensor Data” context of this study.

**Figure 4 sensors-25-07194-f004:**
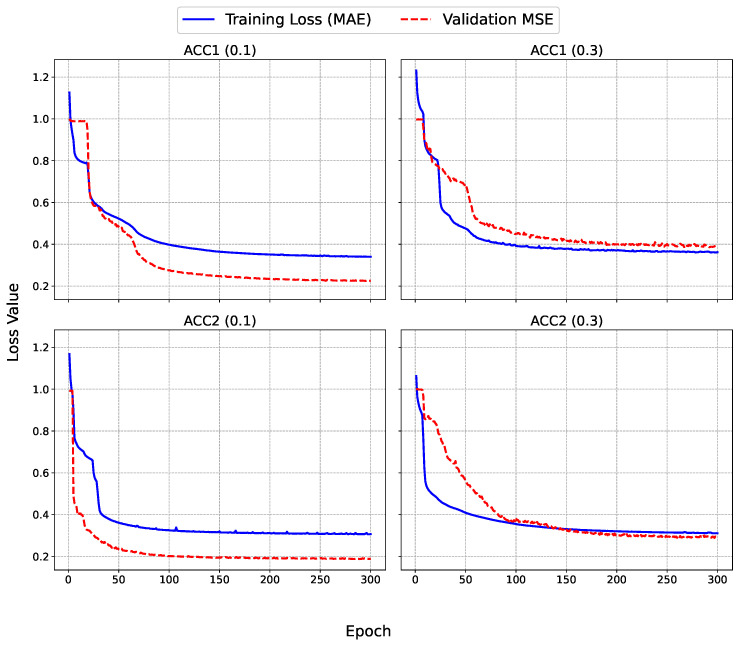
Learning curves for SAITS training on (**top**) ACC1 and (**bottom**) ACC2 datasets at missing factors (**left**) 0.1 and (**right**) 0.3, showing training MAE (blue) and validation MSE (red).

**Figure 5 sensors-25-07194-f005:**
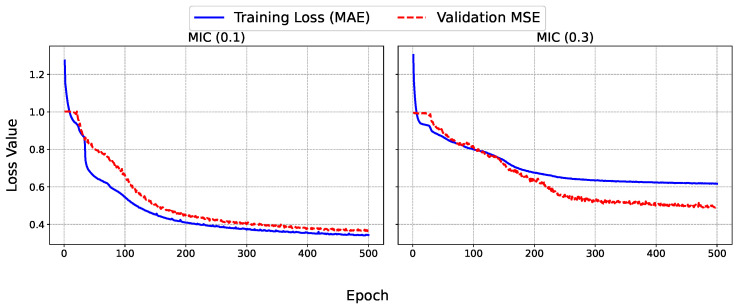
Learning curves for SAITS training on MIC sensor data at missing factors (**left**) 0.1 and (**right**) 0.3, showing training MAE (blue) and validation MSE (red).

**Figure 6 sensors-25-07194-f006:**
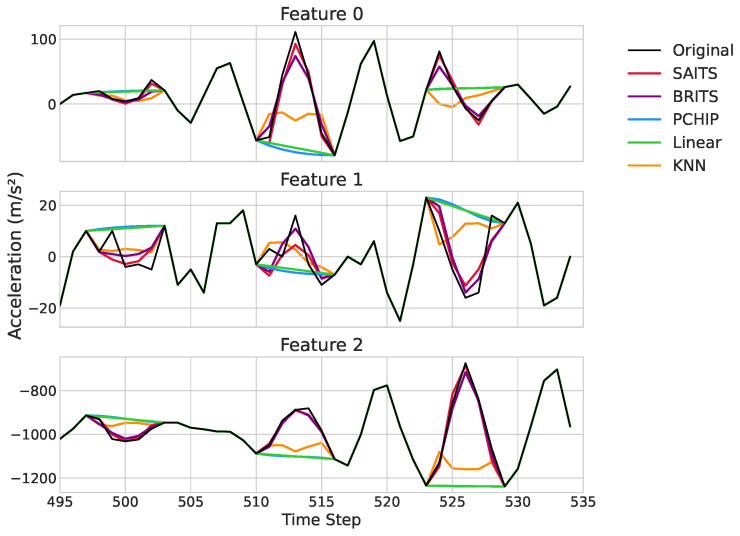
Qualitative comparison of imputation performance for a selected time window from the ACC2 dataset.

**Table 1 sensors-25-07194-t001:** Time-Domain Evaluation Metrics.

Metric	Equation	Description
MAE	$\mathrm{MAE}=\frac{1}{N_{\mathrm{imp}}}\sum_{i\in I}\left|s_{i}-{\hat{s}}_{i}\right|$	Mean Absolute Error quantifies the average absolute deviation between *s* and $\hat{s}$ over imputed indices.
MRE	$\mathrm{MRE}=\frac{1}{N_{\mathrm{imp}}}\sum_{i\in I}\left|\frac{s_{i}-{\hat{s}}_{i}}{s_{i}}\right|$	Mean Relative Error measures the normalized relative discrepancy with respect to the original signal magnitude. Values where $s_{i}=0$ are excluded from computation.
RMSE	$\mathrm{RMSE}=\sqrt{\frac{1}{N_{\mathrm{imp}}}\sum_{i\in I}{\left(s_{i}-{\hat{s}}_{i}\right)}^{2}}$	Root Mean Square Error provides a quadratic penalty that emphasizes larger reconstruction deviations.
Sim	$\mathrm{Sim}=\frac{\sum_{i\in I}\left(s_{i}-\bar{s}\right)\left({\hat{s}}_{i}-\bar{\hat{s}}\right)}{\sqrt{\sum_{i\in I}{\left(s_{i}-\bar{s}\right)}^{2}}\sqrt{\sum_{i\in I}{\left({\hat{s}}_{i}-\bar{\hat{s}}\right)}^{2}}}$	Similarity coefficient (Pearson correlation) quantifies the linear correspondence between original and reconstructed signals, ranging from −1 to 1.
FSD	$\mathrm{FSD}=\frac{\mathrm{RMSE}}{\sqrt{\frac{1}{N_{\mathrm{imp}}}\sum_{i\in I}{\left(s_{i}-\bar{s}\right)}^{2}}}$	Fraction of Standard Deviation provides a dimensionless, scale-invariant error metric normalized by the standard deviation of the original signal.

Note: The MRE calculation excludes cases where the original signal $s_{i}=0$ to avoid division-by-zero. These cases were rare (occurring in <1% of data points) and their exclusion does not affect the comparability of the metric across different methods.

**Table 2 sensors-25-07194-t002:** Frequency-Domain Evaluation Metrics.

Metric	Equation	Description
${\mathrm{RMSE}}_{F}$	${\mathrm{RMSE}}_{F}=\sqrt{\frac{1}{K}\sum_{k=1}^{K}{\left|S_{k}-{\hat{S}}_{k}\right|}^{2}}$	Divergence between spectral coefficients of ground-truth and reconstructed signals across all frequency bins.
${\mathrm{RMSE}}_{F_{\mathrm{Low}}}$	Same as ${\mathrm{RMSE}}_{F}$, computed for low-frequency band (0–15 Hz)	Evaluates imputation fidelity in low frequencies, critical for capturing fundamental dynamics such as touchdown events.
${\mathrm{RMSE}}_{F_{\mathrm{High}}}$	Same as ${\mathrm{RMSE}}_{F}$, computed for high-frequency band (>150 Hz)	Measures performance in high-frequency spectrum, associated with structural phenomena like crack propagation.

**Table 3 sensors-25-07194-t003:** Imputation performance for ACC1 dataset (multivariate) at 0.1 missing factor.

Metric	SAITS	BRITS	PCHIP	Linear	KNN
Feature 0					
MAE	4.31	5.22	23.31	22.17	19.65
MRE	0.66	0.69	2.33	2.15	1.55
RMSE	5.47	6.72	35.74	31.70	27.28
Sim	0.96	0.95	−0.15	−0.17	−0.46
FSD	0.26	0.32	1.69	1.48	1.27
${\mathrm{RMSE}}_{F}$	38.16	46.97	284.62	246.95	201.07
${\mathrm{RMSE}}_{F_{\mathrm{Low}}}$	39.07	37.56	497.69	420.91	199.77
${\mathrm{RMSE}}_{F_{\mathrm{High}}}$	38.11	48.09	172.31	153.28	169.46
Feature 1					
MAE	3.45	3.79	6.53	5.97	5.30
MRE	0.79	0.87	1.49	1.36	1.21
RMSE	4.29	4.69	9.32	7.56	6.67
Sim	0.69	0.64	0.06	0.06	0.05
FSD	0.72	0.79	1.57	1.26	1.11
${\mathrm{RMSE}}_{F}$	29.92	32.77	66.57	52.12	45.99
${\mathrm{RMSE}}_{F_{\mathrm{Low}}}$	38.34	41.84	109.57	80.09	52.70
${\mathrm{RMSE}}_{F_{\mathrm{High}}}$	27.45	30.43	48.73	41.13	41.84
Feature 2					
MAE	13.31	16.16	75.35	70.57	63.85
MRE	0.01	0.02	0.08	0.07	0.07
RMSE	16.97	20.84	113.56	92.99	83.38
Sim	0.96	0.95	−0.15	−0.17	−0.38
FSD	0.27	0.33	1.79	1.44	1.29
${\mathrm{RMSE}}_{F}$	118.17	145.64	906.70	715.55	616.01
${\mathrm{RMSE}}_{F_{\mathrm{Low}}}$	121.40	107.00	1622.79	1282.10	805.98
${\mathrm{RMSE}}_{F_{\mathrm{High}}}$	118.25	152.07	595.38	481.49	511.72

**Table 4 sensors-25-07194-t004:** Imputation performance for ACC1 dataset (multivariate) at 0.3 missing factor.

Metric	SAITS	BRITS	PCHIP	Linear	KNN
Feature 0					
MAE	6.84	7.21	41.22	19.91	16.08
MRE	0.90	0.76	5.52	2.02	1.28
RMSE	9.60	9.70	146.37	28.51	22.49
Sim	0.89	0.92	−0.05	−0.05	0.06
FSD	0.45	0.46	6.78	1.33	1.04
${\mathrm{RMSE}}_{F}$	112.48	121.36	2437.23	390.26	285.68
${\mathrm{RMSE}}_{F_{\mathrm{Low}}}$	106.64	69.93	7211.42	1129.22	449.54
${\mathrm{RMSE}}_{F_{\mathrm{High}}}$	111.43	129.64	1065.47	298.61	285.67
Feature 1					
MAE	3.88	4.18	13.19	6.07	5.16
MRE	0.88	0.96	3.26	1.38	1.18
RMSE	4.85	5.17	47.34	7.67	6.47
Sim	0.59	0.59	0.04	0.05	0.11
FSD	0.81	0.86	8.05	1.28	1.08
${\mathrm{RMSE}}_{F}$	57.68	62.88	777.61	92.93	77.35
${\mathrm{RMSE}}_{F_{\mathrm{Low}}}$	91.52	95.62	2277.47	262.69	144.12
${\mathrm{RMSE}}_{F_{\mathrm{High}}}$	52.06	58.80	333.64	73.01	70.49
Feature 2					
MAE	23.01	24.63	191.88	63.74	53.49
MRE	0.02	0.03	0.20	0.07	0.05
RMSE	31.30	31.94	796.59	84.52	70.35
Sim	0.87	0.91	−0.05	−0.05	−0.07
FSD	0.49	0.49	12.53	1.30	1.08
${\mathrm{RMSE}}_{F}$	367.51	399.99	13,498.91	1159.84	900.88
${\mathrm{RMSE}}_{F_{\mathrm{Low}}}$	328.45	192.13	39,732.57	3078.54	1284.10
${\mathrm{RMSE}}_{F_{\mathrm{High}}}$	379.15	429.74	5548.17	885.36	882.57

**Table 5 sensors-25-07194-t005:** Imputation performance for ACC2 dataset (multivariate) at 0.1 missing factor.

Metric	SAITS	BRITS	PCHIP	Linear	KNN
Feature 0					
MAE	4.81	5.37	27.38	25.73	22.51
MRE	0.59	0.59	2.55	2.32	1.80
RMSE	6.14	6.90	41.20	35.66	30.64
Sim	0.96	0.96	−0.15	−0.17	−0.45
FSD	0.26	0.29	1.74	1.48	1.27
${\mathrm{RMSE}}_{F}$	42.67	48.38	332.85	280.76	227.22
${\mathrm{RMSE}}_{F_{\mathrm{Low}}}$	39.75	36.52	592.70	483.73	232.49
${\mathrm{RMSE}}_{F_{\mathrm{High}}}$	43.33	50.03	198.73	173.14	189.96
Feature 1					
MAE	5.83	6.15	13.03	11.72	10.34
MRE	1.08	1.04	2.34	1.99	1.51
RMSE	7.24	7.67	19.78	14.92	13.13
Sim	0.74	0.71	−0.07	−0.08	−0.21
FSD	0.67	0.71	1.84	1.38	1.21
${\mathrm{RMSE}}_{F}$	50.52	53.55	146.61	103.79	90.80
${\mathrm{RMSE}}_{F_{\mathrm{Low}}}$	53.85	54.82	249.04	165.88	104.55
${\mathrm{RMSE}}_{F_{\mathrm{High}}}$	50.04	53.21	101.91	77.95	81.00
Feature 2					
MAE	14.72	15.95	109.63	106.32	102.91
MRE	0.02	0.02	0.11	0.11	0.11
RMSE	18.79	20.27	160.01	144.79	136.68
Sim	0.98	0.98	−0.12	−0.15	−0.48
FSD	0.19	0.21	1.63	1.47	1.39
${\mathrm{RMSE}}_{F}$	130.33	142.19	1308.62	1163.23	1053.91
${\mathrm{RMSE}}_{F_{\mathrm{Low}}}$	153.14	131.85	2560.79	2271.34	1559.17
${\mathrm{RMSE}}_{F_{\mathrm{High}}}$	120.50	131.58	693.65	611.65	740.76

**Table 6 sensors-25-07194-t006:** Imputation performance for ACC2 dataset (multivariate) at 0.3 missing factor.

Metric	SAITS	BRITS	PCHIP	Linear	KNN
Feature 0					
MAE	6.97	7.83	41.94	23.32	18.63
MRE	0.80	0.73	5.06	2.18	1.51
RMSE	9.28	10.52	135.07	32.21	25.54
Sim	0.92	0.93	−0.05	−0.05	0.04
FSD	0.39	0.44	5.77	1.33	1.05
${\mathrm{RMSE}}_{F}$	109.75	132.34	2233.13	447.27	326.43
${\mathrm{RMSE}}_{F_{\mathrm{Low}}}$	72.82	68.05	6567.73	1305.50	501.03
${\mathrm{RMSE}}_{F_{\mathrm{High}}}$	116.62	142.55	1005.86	336.24	323.42
Feature 1					
MAE	6.45	6.36	27.12	11.11	9.28
MRE	1.06	1.01	5.24	1.89	1.33
RMSE	8.10	7.97	103.14	14.13	11.79
Sim	0.67	0.69	−0.02	−0.02	0.01
FSD	0.75	0.74	9.63	1.30	1.08
${\mathrm{RMSE}}_{F}$	96.85	96.17	1717.24	175.10	141.61
${\mathrm{RMSE}}_{F_{\mathrm{Low}}}$	107.34	105.38	5046.10	476.64	223.93
${\mathrm{RMSE}}_{F_{\mathrm{High}}}$	96.63	96.40	722.50	139.08	135.66
Feature 2					
MAE	27.61	23.75	150.36	100.42	84.09
MRE	0.03	0.02	0.16	0.10	0.09
RMSE	37.01	30.69	404.41	134.66	111.43
Sim	0.92	0.96	−0.05	−0.05	−0.12
FSD	0.38	0.31	4.27	1.36	1.12
${\mathrm{RMSE}}_{F}$	438.42	385.43	6596.02	1923.61	1462.76
${\mathrm{RMSE}}_{F_{\mathrm{Low}}}$	243.45	222.87	19,606.04	5897.15	2257.65
${\mathrm{RMSE}}_{F_{\mathrm{High}}}$	412.46	350.17	2872.99	1128.70	1136.01

**Table 7 sensors-25-07194-t007:** Imputation performance for MIC sensor data (single feature) at 0.1 missing factor.

Metric	SAITS	BRITS	PCHIP	Linear	KNN
MAE	712.78	1001.92	2028.70	1900.55	1820.80
MRE	0.25	0.26	0.48	0.61	0.78
RMSE	1136.47	1415.63	3154.64	2564.66	2450.31
Sim	0.80	0.63	0.09	0.08	0.00
FSD	0.59	0.77	1.57	1.24	1.14
${\mathrm{RMSE}}_{F}$	7920.46	9913.73	22,956.49	17,917.50	17,091.78
${\mathrm{RMSE}}_{F_{\mathrm{Low}}}$	9834.48	12,212.87	39,463.54	30,553.30	24,433.59
${\mathrm{RMSE}}_{F_{\mathrm{High}}}$	7138.40	9031.52	15,860.31	12,692.10	13,566.03

**Table 8 sensors-25-07194-t008:** Imputation performance for MIC sensor data (single feature) at 0.3 missing factor.

Metric	SAITS	BRITS	PCHIP	Linear	KNN
MAE	976.14	1336.50	3712.99	1961.64	1782.02
MRE	0.96	1.28	3.63	2.01	1.88
RMSE	1507.34	1854.87	12,120.99	2694.22	2497.36
Sim	0.73	0.50	0.05	0.06	0.05
FSD	0.69	0.89	5.35	1.24	1.11
${\mathrm{RMSE}}_{F}$	17,747.03	22,914.54	195,058.93	33,205.95	30,488.47
${\mathrm{RMSE}}_{F_{\mathrm{Low}}}$	27,323.59	32,388.56	569,316.78	87,479.48	65,995.89
${\mathrm{RMSE}}_{F_{\mathrm{High}}}$	14,799.94	19,779.72	82,871.07	22,382.76	22,480.07

## Data Availability

The code (version 1.0) is available in the following GitHub repository: [https://github.com/MehdiBejani/SAITS-Sensor-Imputation/tree/main]. (accessed on 10 October 2025). Data is unavailable due to commercial restrictions.

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
