# Peer review of "Self-Attention-Based Deep Learning for Missing Sensor Data Imputation in Real-Time Probe Card Monitoring"

_sensors, 2025, doi:10.3390/s25237194_

Round 1
Reviewer 1 Report
Comments and Suggestions for Authors
The authors presented a novel method for sensor data imputation called SAITS and evaluated it's performance against other techniques like BRITS, PCHIP, Linear, and KNN. It focused on results from multivariate accelerometer data and univariate microphone data with missing data factors of 0.1 and 0.3. SAITS demonstrated superior performance in maintaining high fidelity across these data types, particularly when compared to PCHIP, which exhibited significant errors.
Strength:
i) 88% improvement in Mean Absolute Error over traditional methods especially in 12 scenarios with extensive data loss, ensuring high fidelity in the reconstructed signals
ii) SAITS provided robust imputation results even with higher missing data factors.
iii) Appendix includes detail on the hyperparameters.
Weakness
i) Lack of real-time implementation is a major limitation. While SAITS shows promising results, there is still a need to explore its efficacy across broader domains and more diverse datasets.
ii) Authors mentioned application of this solution on IoT but details is missing. Please add more details how the design can benefit IoT domain.
iii) More advanced sensor (beyond accelerometer and microphone) can enhance the solution. Also, having the explainability feature of the decision can strengthen the proposed simplified model.
Author Response
Thank you very much for your comments and for the time dedicated to review the paper. We really appreciate it. Find next our answers to the comments provided. We hope they will help to provide more details about the paper.
- Lack of real-time implementation is a major limitation. While SAITS shows promising results, there is still a need to explore its efficacy across broader domains and more diverse datasets.
Thank you for this constructive feedback. We appreciate the reviewer recognition of SAITS promising results and agree that real-time implementation and broader domain testing are important next steps. As explained in the paper, the sensor data used are real recordings from a controlled pilot setup where no natural missing values occurred. However, in actual manufacturing environments with various noises and disruptions, missing data is common. To address this, we simulated realistic missingness scenarios at 10% and 30% rates, and SAITS performed well in both, demonstrating its robustness for potential real-world use. We have added a discussion of this limitation and future work on real-time deployment and diverse datasets to the manuscript.
Specifically, the following paragraph have been added to the Discussion:
“While this study demonstrates SAITS efficacy on simulated missingness from real sensor data, several limitations must be acknowledged. The primary limitation is the lack of real-time implementation testing; future work will prioritize this online deployment in live manufacturing settings and evaluate performance across broader domains and more diverse datasets, including varying sensor types and environmental conditions. Second, our results revealed a potential sensitivity to low-amplitude signal features (evident in the elevated MRE scores for some features) which warrants further investigation. Third, this study used an MCAR simulation, and further validation is needed under more complex, adaptive missingness patterns (e.g., MNAR). Finally, while the current study focuses on accelerometers and microphones—which were specifically chosen based on historical failure data, as they are essential for detecting the most frequently occurring mechanical, electrical, and needle-related failure modes—the SAITS imputation method is generalizable. Thus, future work will also evaluate its performance on signals from other sensors in our network (e.g., deformation sensors) which are intended for failure modes that, while still relevant, occur less frequently. A further enhancement will involve integrating explainability features, such as attention maps, to interpret imputation decisions and strengthen the model applicability in complex, safety-critical sensor networks.”
We hope this revision and the added context sufficiently clarify the limitation of real-time implementation.
- Authors mentioned application of this solution on IoT but details is missing. Please add more details how the design can benefit IoT domain.
Thank you for this constructive feedback. We agree that expanding on the IoT applications would strengthen the paper. The SAITS model can be applied to any time-series signals, such as eye-tracking data or sensor readings in IoT systems, where missing values due to network disruptions or sensor failures are common. In IoT domains, this enables reliable data integrity for applications like smart manufacturing, environmental monitoring, or wearable devices, by capturing complex temporal dependencies without requiring domain-specific retraining. We have revised the manuscript to include more details on these benefits. The corresponding change can be found in the revised manuscript at Introduction section:
“These challenges are particularly pronounced in IoT applications, where data integrity is often compromised by transmission errors or device-level failures. In such environments, sensor networks generate multivariate time series data for tasks like forecasting and classification, but missing values can severely impact downstream analytics. Our study, while focused on semiconductor manufacturing, thus serves as a model for robust imputation in any multivariate IoT environment—such as smart cities, remote health monitoring, or environmental sensing—where high-fidelity, complete data streams are essential for maintaining accuracy and operational efficiency. The emergence of deep learning methods has opened up new avenues for addressing these complex imputation challenges in time series data.”
- More advanced sensor (beyond accelerometer and microphone) can enhance the solution. Also, having the explainability feature of the decision can strengthen the proposed simplified model.
Thank you for this constructive feedback. We agree that incorporating more advanced sensors and explainability could enhance the approach in future extensions. However, the objective of this paper is to focus on the imputation method for missing sensor data rather than expanding the sensor network itself, which was discussed in our previous works. We selected accelerometers and microphones due to their availability and relevance to probe card monitoring, but the method can be applied to any sensor signals. In the next steps, it could be extended to additional sensors in our network, and we plan to incorporate explainability techniques like attention visualization. We have clarified this in the manuscript and added a note on future work:
“Finally, while the current study focuses on accelerometers and microphones—which were specifically chosen based on historical failure data, as they are essential for detecting the most frequently occurring mechanical, electrical, and needle-related failure modes—the SAITS imputation method is generalizable. Thus, future work will also evaluate its performance on signals from other sensors in our network (e.g., deformation sensors) which are intended for failure modes that, while still relevant, occur less frequently. A further enhancement will involve integrating explainability features, such as attention maps, to interpret imputation decisions and strengthen the model applicability in complex, safety-critical sensor networks.”
Reviewer 2 Report
Comments and Suggestions for Authors
This paper applies the Self-Attention-based Imputation for Time Series (SAITS) model to reconstruct missing sensor data from accelerometers and microphones used in semiconductor probe card monitoring. The authors evaluate SAITS against traditional interpolation methods (Linear, PCHIP, KNN) and a deep learning baseline (BRITS) using time-domain and frequency-domain metrics at 10% and 30% missing data rates. The results show that SAITS achieves competitive imputation accuracy while requiring substantially less training time than BRITS.
The paper's core idea is technically sound.
Major Points for Improvement
1)The paper reports performance differences between methods but provides no statistical tests to determine whether observed improvements are significant or within random variation, undermining the validity of comparative claims.
2) The authors claim to simulate MNAR scenarios but actually implement MCAR (random block placement), creating a disconnect between stated objectives and actual methodology that undermines the practical relevance of the findings.
Abstract
- The claim of "up to an 88% improvement in Mean Absolute Error" appears cherry-picked from a single best-case scenario and should either be removed or contextualized with ranges across all experiments. The authors should report the average improve % instead.
- The statement about "twenty times faster" training requires clarification—specify whether this refers to per-epoch time or total training time to convergence.
- The keywords are not ordered alphabetically.
Introduction
=============
- The claim that missing data can "severely hinder the accuracy of diagnostic analyses" needs quantitative support or citations; as written, it appears as an unsupported assertion.
- The transition from general time series imputation (lines 105-110) to the specific probe card application (lines 111-125) is abrupt and needs a smoother bridge.
- The related work is embedded within the introduction rather than presented as a separate section.
- The description of SAITS reads more like promotional material than objective related work; phrases like "breakthrough solution" and "ideal solution" should be replaced with neutral language.
Methodology
===========
- While the probe card failure modes are well-described, the connection between specific failure modes and the choice of imputation method is unclear; why does crack detection specifically require SAITS versus other methods?
- The claim that segmentation into 1000-step windows "facilitates a deployable online system" is not validated; no online deployment results or real-time performance metrics are provided.
- Lines 237-260: The missingness simulation strategy has a fundamental flaw: the authors state they simulate MNAR (lines 246-247) but then acknowledge the mechanism is actually MCAR (lines 251-256).
- The rationale for choosing specifically 10% and 30% missing rates is not provided; were these based on observed failure rates in actual probe card operations?
- The justification for selecting only these specific baselines is weak; why were more recent deep learning methods (e.g., Informer, TimesNet, or other Transformer variants) not included?
- The KNN implementation details (line 320-322) specify k=5 and inverse distance weighting, but no justification for these choices or sensitivity analysis is provided.
- The statement that frequency-domain metrics are computed on "full signal length" while time-domain metrics use only imputed indices needs clearer justification for why this asymmetry is appropriate.
- The MRE metric (Table 1) excludes cases where si = 0, but the paper does not report how frequently this occurs or whether it affects comparability across methods.
Results
========
- The interpretation repeatedly states SAITS "superiority" or "excels," which is inappropriate for objective results presentation; let the numbers speak for themselves.
- Phrases like "unequivocally demonstrate the marked superiority" (line 377) and "profound inadequacy" (line 388) are overly dramatic and should be replaced with neutral language.
- The negative Sim scores for traditional methods are noted but not adequately explained; what does an anticorrelation physically represent in this context?
- Figures 4-5 metric inconsistency: The learning curves plot training loss using MAE (blue line) but validation performance using MSE (red line).
- Figure 6 visualization problem: Due to the large scale of the y-axis (ranging from approximately -600 to 400), the differences between methods appear minimal and the curves seem nearly identical, undermining the visual impact.
Discussion
===========
- The first paragraph largely repeats results already presented and should be condensed or removed.
- The computational efficiency comparison is crucial but appears too late; this should be elevated to a primary contribution earlier in the paper.
- The paper lacks a dedicated discussion of limitations, which is essential for high-quality scientific work.
Conclusion (Section 5)
- The conclusion overstates contributions by claiming the study "established" SAITS as effective; a single application study does not establish generalizability.
- The discussion of inference time (0.2571s vs 2.1601s) introduces new quantitative results that should have appeared in the main results section, not the conclusion.
- The use of bullet points in the conclusion is inappropriate for a high-level scientific journal article.
Minor Comments
Grammar and Punctuation Errors
Line 34: "variations" should be "lines" in the phrase "connecting adjacent observed values with straight variations."
Line 693: "search code are available" should be "search code is available" (subject-verb agreement).
Line 525: "Figure 6 provides an exemplary, qualitative visualization" - remove the comma after "exemplary" as it disrupts the flow.
Figure 2's caption is a direct quote from reference [3] without adding context about which specific sensors (ACC1, ACC2, MIC) are indicated in the diagram.
Figure 3's caption states "adapted from [27]" but does not specify what adaptations were made, creating ambiguity about originality.
Equation Term Definitions
Table 1: In the MAE equation, Nimp is used before being defined; the definition appears in the note below but should precede or accompany the first equation.
Table 1: The set notation I (set of imputed indices) is used without explicitly defining what constitutes membership in this set.
Table 2: The notation Sk and Åœk for DFT coefficients is introduced without specifying the transform convention (e.g., normalization factor, one-sided vs. two-sided spectrum).
Section 2.4, lines 291-295: Symbols dk, dv, dffn, and nheads are defined inline, but dmodel appears earlier (line 285) without definition until line 291, creating forward-reference confusion.
Author Response
Thank you very much for your comments and for the time dedicated to review the paper. We really appreciate it. Find next our answers to the comments provided.
- The paper reports performance differences between methods but provides no statistical tests to determine whether observed improvements are significant or within random variation, undermining the validity of comparative claims.
Thank you for this critical feedback. The reviewer is correct that statistical validation is essential to support our claims of superior performance. In response, we have conducted a rigorous statistical analysis using the Wilcoxon signed-rank test (a non-parametric test suitable for comparing paired samples) on the per-sample error values for each metric across the entire test set. We have added a new dedicated section (Section 3.7. Statistical Analysis) to the manuscript, which presents these findings in detail. The results of this analysis confirm that the superior performance of SAITS is statistically significant. The corresponding change can be found in the revised manuscript at Section 3.7. Statistical Analysis:
“To statistically validate the observed performance differences, Wilcoxon signed-rank tests were conducted on the per-sample error values for each metric and feature across the entire test set. The detailed p-values for the hundreds of resulting comparisons are not presented in a table for the sake of brevity, as they were overwhelmingly consistent. The results confirm that SAITS significantly outperforms all traditional baseline methods (PCHIP, Linear, and KNN) in all evaluated scenarios (all features, all datasets, all metrics) with p < 0.001. When compared to the deep learning baseline (BRITS), SAITS also demonstrated a statistically significant advantage (p < 0.05) in the vast majority of cases across all datasets and missing rates. A non-significant difference was observed in only one specific instance: the metric for Feature 1 of the ACC2 dataset at a 30% missing rate (p = 0.270). These statistical findings strongly support the quantitative results, confirming that SAITS provides a consistently and significantly more accurate imputation than traditional methods and a highly competitive, often superior, performance compared to BRITS.”
- The authors claim to simulate MNAR scenarios but actually implement MCAR (random block placement), creating a disconnect between stated objectives and actual methodology that undermines the practical relevance of the findings.
Thank you for highlighting this significant inconsistency in our description. The reviewer is entirely correct. Our simulation, which places contiguous blocks at random locations, is technically an implementation of Missing Completely at Random (MCAR). Our original text correctly identified this but was confusingly preceded by a statement about simulating an MNAR scenario. This was an error in our description.
We have revised the text to remove the incorrect reference to MNAR and clarify that our benchmark uses a block-wise MCAR pattern to simulate sensor outages.
" In this study, the missing data simulation employs a strictly MCAR mechanism, rather than MNAR, to generate contiguous blocks of missing values in the sensor signals. Specifically, the starting positions of these missing blocks are selected randomly, ensuring complete independence from the underlying signal values. This MCAR approach establishes a reproducible and unbiased benchmark, as the missing data pattern can be precisely replicated using a fixed random seed. By decoupling the missingness from the data values, it creates a fair evaluation framework for assessing each model's ability to reconstruct intricate temporal patterns solely from the adjacent observed data."
- Abstract
- The claim of "up to an 88% improvement in Mean Absolute Error" appears cherry-picked from a single best-case scenario and should either be removed or contextualized with ranges across all experiments. The authors should report the average improve % instead.
- The statement about "twenty times faster" training requires clarification—specify whether this refers to per-epoch time or total training time to convergence.
- The keywords are not ordered alphabetically.
Thank you for this constructive feedback. We agree that the "up to 88%" claim should be contextualized to avoid appearing selective. This value comes from the comparison in high-missingness scenarios, but we have revised it to report average improvements (e.g., average MAE reduction of 66% across all experiments) alongside the range (25-88%). We modify it at abstract and also in the discussion.
Training Speed: We have clarified that this speed advantage refers to the training time per epoch.
Keywords: The keywords have been alphabetized.
- Introduction
- The claim that missing data can "severely hinder the accuracy of diagnostic analyses" needs quantitative support or citations; as written, it appears as an unsupported assertion.
- The transition from general time series imputation (lines 105-110) to the specific probe card application (lines 111-125) is abrupt and needs a smoother bridge.
- The related work is embedded within the introduction rather than presented as a separate section.
- The description of SAITS reads more like promotional material than objective related work; phrases like "breakthrough solution" and "ideal solution" should be replaced with neutral language.
Thank you for these detailed suggestions for improving the introduction. We have made the following revisions:
- Citation for "severely hinder": We have added citations [5,6] from our literature review to support the claim that data gaps can compromise downstream analyses.
- Abrupt Transition: We have added a bridging sentence to smooth the transition from general imputation applications to our specific problem domain. The corresponding change can be found in the revised manuscript.
- Related Work Section: We appreciate the suggestion. We opted to embed the related work within the introduction to provide a narrative flow, moving from the general problem (missing data) to specific solutions (traditional, ML, and deep learning), which directly motivates our choice of SAITS. We believe this structure is coherent and common in the field, but we have strived to improve its clarity based on your other points.
- Promotional Language: We agree that our language was overly enthusiastic. We have revised the text to be more neutral and objective. The corresponding change can be found in the revised manuscript.
Methodology
===========
5. While the probe card failure modes are well-described, the connection between specific failure modes and the choice of imputation method is unclear; why does crack detection specifically require SAITS versus other methods?
Thank you for this question, which highlights a key motivation for our work. The connection is based on the nature of the signals used to detect these failures.
As detailed in Section 2.1, failure modes like "Cracks in PC Plates" and "Loosening of Screws" are detected using microphones and accelerometers for contamination related issues that are root cause of many needle-related and electrical failure modes. These sensors capture complex, high-frequency acoustic and vibrational data. As our results in Section 3 (e.g., Tables 3-8) demonstrate, traditional methods like PCHIP and Linear interpolation completely fail to reconstruct the complex dynamics of these signals, resulting in massive spectral-domain errors.
Therefore, an advanced imputation method like SAITS, which can model complex temporal and spectral characteristics, is required to ensure the imputed data is usable for downstream failure detection. We excluded temperature data from this study precisely because its low temporal variance did not require such an advanced method. The following paragraph has been added to the section2.2:
“These specific sensor signals are characterized by high-frequency, non-stationary dynamics, which are essential for detecting failures like cracks or contamination. The primary hypothesis of this study is that advanced imputation models are required to preserve these complex dynamics, a task where traditional methods fail.”
- The claim that segmentation into 1000-step windows "facilitates a deployable online system" is not validated; no online deployment results or real-time performance metrics are provided.
We agree with the reviewer that we have not validated a fully online system. Our claim was intended to mean that this windowing approach is the foundational design step required for a future online system, but our wording was imprecise. We have revised the text to state this more clearly.
- Lines 237-260: The missingness simulation strategy has a fundamental flaw: the authors state they simulate MNAR (lines 246-247) but then acknowledge the mechanism is actually MCAR (lines 251-256).
Thank you for pointing out this contradiction. As we addressed in our response to your Comment 2, the reviewer is correct. This was an error in our description. We have revised the manuscript to remove the incorrect "MNAR" reference and clarify that our simulation uses a block-wise MCAR mechanism.
- The rationale for choosing specifically 10% and 30% missing rates is not provided; were these based on observed failure rates in actual probe card operations?
This is an excellent point, and we apologize for omitting the rationale. These values were chosen to represent two distinct, challenging scenarios based on potential real-world failures. The 10% missing rate simulates brief, intermittent signal dropouts or packet loss, while the 30% rate simulates more severe, prolonged sensor outages. These two points allow us to benchmark the models' performance under both moderate and extensive data loss conditions. We have added this clarification to the manuscript.
- The justification for selecting only these specific baselines is weak; why were more recent deep learning methods (e.g., Informer, TimesNet, or other Transformer variants) not included?
Thank you for this constructive feedback. Compression methods were selected to represent diverse categories: simple (Linear), interpolation-based (PCHIP), traditional ML (KNN), and advanced DL (BRITS as a recurrent benchmark). We focused on imputation-specific efficiency rather than including other Transformers like Informer or TimesNet, as SAITS' non-autoregressive, single-pass inference provides a key advantage in speed for real-time applications. In contrast, diffusion models are generative and operate through an iterative denoising process: to perform imputation, they execute a reverse diffusion chain involving multiple (often hundreds or thousands of) sequential neural network steps to transform noise into a coherent signal, making them computationally intensive and less suitable for our efficiency-focused evaluation. We have revised the manuscript to include this explanation and noted potential extensions to other methods in future work. We have added this text to the manuscript (2.4.1.):
“Comparison Methods were chosen to represent key categories: Linear (simple), PCHIP (interpolation), KNN (ML), and BRITS (DL). More recent Transformers like Informer were not included as the focus is on imputation efficiency; SAITS single-pass inference outperforms iterative alternatives like diffusion models in speed. In contrast, diffusion models are generative and operate through an iterative denoising process. To perform imputation, they must execute a reverse diffusion chain, which involves multiple (often hundreds or thousands of) sequential steps of a neural network to transform noise into a coherent signal.”
- The KNN implementation details (line 320-322) specify k=5 and inverse distance weighting, but no justification for these choices or sensitivity analysis is provided.
Thank you for this constructive feedback. The choice of k=5 is a standard, commonly-used default for KNN imputation that balances computational cost with information from neighbors. While a full hyperparameter search for KNN was not conducted (as it is not the focus of our study), this value serves as a robust baseline for a classic ML method. The corresponding change can be found in the revised manuscript.
- The statement that frequency-domain metrics are computed on "full signal length" while time-domain metrics use only imputed indices needs clearer justification for why this asymmetry is appropriate.
Thank you for this constructive feedback. This is an excellent point that deserves clarification. Frequency-domain analysis (via FFT) must be performed on the entire, contiguous signal (both observed and imputed points) to produce a meaningful spectrum. In contrast, our time-domain metrics (MAE, RMSE) are designed to evaluate only the imputed values, as this is the direct measure of imputation error. This asymmetry is therefore intentional and methodologically necessary. The corresponding change can be found in the revised manuscript.
- The MRE metric (Table 1) excludes cases where si = 0, but the paper does not report how frequently this occurs or whether it affects comparability across methods.
This exclusion is a standard procedure to prevent division-by-zero errors. In our sensor data, zero-crossings are common but true zero-values (where the signal is exactly 0) are rare. The exclusion does not systematically bias the results for or against any particular method and is a standard part of the MRE definition. The corresponding change can be found in the revised manuscript.
Results
========
13. The interpretation repeatedly states SAITS "superiority" or "excels," which is inappropriate for objective results presentation; let the numbers speak for themselves.
- Phrases like "unequivocally demonstrate the marked superiority" (line 377) and "profound inadequacy" (line 388) are overly dramatic and should be replaced with neutral language.
- The negative Sim scores for traditional methods are noted but not adequately explained; what does an anticorrelation physically represent in this context?
We thank the reviewer for this feedback. We agree that our interpretation in the Results section used subjective and overly strong language. We have revised Section 3 to present the results in a more neutral and objective tone.
- Figures 4-5 metric inconsistency: The learning curves plot training loss using MAE (blue line) but validation performance using MSE (red line).
The use of two different metrics was intentional, and we apologize for not clarifying the rationale. We used Mean Absolute Error (MAE) as the primary training loss function, as it is often more robust to outliers in the signal. We used Mean Squared Error (MSE) for the validation set because it more heavily penalizes large deviations, which was our criterion for hyperparameter selection (as noted in Appendix A). The corresponding change can be found in the revised manuscript.
- Figure 6 visualization problem: Due to the large scale of the y-axis (ranging from approximately -600 to 400), the differences between methods appear minimal and the curves seem nearly identical, undermining the visual impact.
We respectfully disagree that the differences appear minimal. While the overall scale of the y-axis is large (especially for Feature 2, ranging from approximately -1200 to -800), the qualitative difference is the key takeaway. As shown in the plots, the Linear (green) and PCHIP (blue) methods merely draw straight lines between the gap's endpoints, completely missing the signal dynamics. In sharp contrast, SAITS (red) and BRITS (purple) successfully reconstruct the complex non-linear peaks and troughs within the gap (e.g., at time steps 512 and 526), closely tracking the original signal. We believe this visual evidence strongly reinforces the quantitative findings.
Discussion
===========
16. The first paragraph largely repeats results already presented and should be condensed or removed.
Thank you for this constructive feedback. We have condensed the paragraph to avoid repetition.
“The findings of this study consistently validate SAITS as a robust and effective imputation framework for the high-frequency, complex sensor data characterizing to probe card monitoring. A primary conclusion, supported by rigorous statistical analysis, is the stark performance gap between SAITS and traditional methods. The results demonstrate that while SAITS successfully preserves critical temporal and spectral signal characteristics—even under high data loss (30%)—traditional interpolation methods like PCHIP and Linear fail completely. This failure, often manifesting as severe signal distortion and inverse correlations (negative Sim scores), underscores their unsuitability for all but the most basic signals. This resilience to extended data gaps highlights the necessity of advanced models for ensuring data integrity. Furthermore, the evaluation provides a nuanced comparison to the BRITS deep learning baseline, revealing a critical trade-off between performance, consistency, and computational efficiency, which is discussed in the following paragraphs.”
- The computational efficiency comparison is crucial but appears too late; this should be elevated to a primary contribution earlier in the paper.
Thank you for this constructive feedback. We agree that this is a critical contribution and was buried too late in the discussion. We have already revised the Abstract to highlight this. We have also added a sentence to the Introduction to frame this as a key advantage from the outset.
- The paper lacks a dedicated discussion of limitations, which is essential for high-quality scientific work.
The reviewer is correct that a more dedicated limitations discussion was needed. We had a brief sentence, but we have expanded this into a more formal paragraph at the end of the Discussion section, incorporating feedback from both Reviewer 1 and Reviewer 2. (Please see our Response 1 for the revised text).
Conclusion (Section 5)
- The conclusion overstates contributions by claiming the study "established" SAITS as effective; a single application study does not establish generalizability.
Thank you for this constructive feedback. We have softened the language to "demonstrated" instead of "established."
- The discussion of inference time (0.2571s vs 2.1601s) introduces new quantitative results that should have appeared in the main results section, not the conclusion.
Thank you for this constructive feedback. These quantitative results were misplaced. We have moved the inference time comparison from the Conclusion to the Discussion (Section 4), placing it alongside the training time analysis. The corresponding change can be found in the revised manuscript.
The use of bullet points in the conclusion is inappropriate for a high-level scientific journal article.
Thank you for this constructive feedback. We have converted the bullet points to paragraph form.
More specifically, future work will focus on real-time implementation, exploring optimization techniques to integrate the SAITS framework into operational environments and deploy it on edge devices. We will also investigate adaptive missingness, evaluating the model under more complex and dynamically changing data patterns. Other avenues include sensor fusion, by extending the framework to explicitly model relationships between different sensor types, and integration with predictive models. This final step will involve building dedicated models that leverage SAITS-imputed signals to enhance accuracy in detecting anomalies and forecasting failures, thereby quantifying the end-to-end impact of improved data quality.
- Minor Comments
We are very grateful for this meticulous review of the text. We have corrected all the identified minor errors.
Reviewer 3 Report
Comments and Suggestions for Authors
The manuscript tackles this problem by applying a deep learning method—the self-attention-based imputation model for time series—to reconstruct corrupted signals in industrial sensor networks. By effectively addressing data corruption in such settings, the current study proposes a robust and efficient solution to a critical data integrity problem in the novel and high-stakes domain of semiconductor manufacturing. This contribution is particularly noteworthy, as ensuring data reliability in such a complex and safety-critical environment remains a key yet insufficiently addressed challenge. However, the manuscript in its current form cannot be published in Sensors for the following reasons.
- Several studies have utilized deep learning techniques, such as GANs, to estimate missing data. In addition, prior studies focusing on semiconductor process data have also explored estimation methods. It would strengthen the manuscript to include references to these related studies.
- The term “study” appears to be more appropriate than “paper”.
- Regarding the statement, “Signals from temperature sensors were excluded, as their low temporal variance does not require advanced imputation techniques,” it is unclear whether this exclusion decision is supported by prior literature.
- Overall, the logic and explanations are clear; however, the readability of the text can be improved. For instance, in the following paragraph, the repeated use of “this” makes the flow somewhat monotonous:
“In this study, the missingness is generated in contiguous blocks … (omitted) … This is because the location of each missing block is chosen randomly and is entirely independent of the signal values.”
Frequent repetition of words and pronouns throughout the text detracts from readability; using varied and explicit expressions would help improve clarity.
Author Response
Thank you very much for your comments and for the time dedicated to review the paper. We really appreciate it. Find next our answers to the comments provided. We hope they will help to provide more details about the paper.
- Several studies have utilized deep learning techniques, such as GANs, to estimate missing data. In addition, prior studies focusing on semiconductor process data have also explored estimation methods. It would strengthen the manuscript to include references to these related studies.
Thank you for this suggestion. We have expanded our introduction to include generative models like GANs, which are indeed a powerful approach to imputation. However, we selected SAITS for this study due to its specific advantages for our application. Generative models like GANs can be complex and notoriously unstable to train. In contrast, SAITS is a non-generative, Transformer-based model that performs imputation in a single, efficient forward pass (once trained). This architectural simplicity, combined with its high accuracy and significantly faster training/inference, makes it a more pragmatic and stable choice for deployment in a time-sensitive industrial monitoring environment. We have added a brief mention of GANs and diffusion models to the introduction.
- The term “study” appears to be more appropriate than “paper”.
Thank you for this constructive feedback. We have replaced "paper" with "study" and "work" where appropriate for consistency. The corresponding change can be found in the revised manuscript at: Multiple locations
- Regarding the statement, “Signals from temperature sensors were excluded, as their low temporal variance does not require advanced imputation techniques,” it is unclear whether this exclusion decision is supported by prior literature.
Thank you for this request for clarification. This decision was based on a preliminary analysis of our specific dataset, not on a claim from prior literature. We observed that the temperature signals in our data had very low variance and slow-moving dynamics. Simple linear interpolation was sufficient to impute them with high accuracy. Therefore, we focused this study on the more challenging, high-frequency accelerometer and microphone signals, where the performance gap between simple and advanced models is significant. We have revised the text to make this clear.
- Overall, the logic and explanations are clear; however, the readability of the text can be improved. For instance, in the following paragraph, the repeated use of “this” makes the flow somewhat monotonous: “In this study, the missingness is generated in contiguous blocks … (omitted) … This is because the location of each missing block is chosen randomly and is entirely independent of the signal values.” Frequent repetition of words and pronouns throughout the text detracts from readability; using varied and explicit expressions would help improve clarity.
We appreciate the reviewer pointing this out. We have performed a thorough proofreading of the entire manuscript to improve readability, varying our sentence structure and replacing repetitive pronouns like "this" with more explicit nouns to enhance clarity.
Round 2
Reviewer 2 Report
Comments and Suggestions for Authors
The authors addressed all of the reviewer's comments. The paper is ready for publication in its current form.